# The Clinical Usability Evaluation of an Attachable Video Laryngoscope in the Simulated Tracheal Intubation Scenario: A Manikin Study

**DOI:** 10.3390/bioengineering11060570

**Published:** 2024-06-05

**Authors:** Won-Jun Lee, Hee-Young Lee, Sun-Ju Kim, Kang-Hyun Lee

**Affiliations:** Department of Emergency Medicine, Yonsei University Wonju College of Medicine, Wonju 26426, Gangwon State, Republic of Korea; no1dtxno1@gmail.com (W.-J.L.); hylee3971@yonsei.ac.kr (H.-Y.L.); mirage01240@gmail.com (S.-J.K.)

**Keywords:** video laryngoscope, difficult airway management, tongue edema, Pentax AWS

## Abstract

The aim of this study was to assess the usefulness of an attachable video laryngoscope (AVL) by attaching a camera and a monitor to a conventional Macintosh laryngoscope (CML). Normal and tongue edema airway scenarios were simulated using a manikin. Twenty physicians performed tracheal intubations using CML, AVL, Pentax Airwayscope^®^ (AWS), and McGrath MAC^®^ (MAC) in each scenario. Ten physicians who had clinical experience in using tracheal intubation were designated as the skilled group, and another ten physicians who were affiliated with other departments and had little clinical experience using tracheal intubation were designated as the unskilled group. The time required for intubation and the success rate were recorded. The degree of difficulty of use and glottic view assessment were scored by participants. All 20 participants successfully completed the study. There was no difference in tracheal intubation success rate and intubation time in the normal airway scenario in both skilled and unskilled groups. In the experienced group, AWS had the highest success rate (100%) in the tongue edema airway scenario, followed by AVL (60%), MAC (60%), and CML (10%) (*p* = 0.001). The time required to intubate using AWS was significantly shorter than that with AVL (10.2 s vs. 19.2 s) or MAC (10.2 s vs. 20.4 s, *p* = 0.007). The difficulty of using AVL was significantly lower than that of CML (7.8 vs. 2.8; *p* < 0.001). For the experienced group, AVL was interpreted as being inferior to AWS but better than MAC. Similarly, in the unskilled group, AVL had a similar success rate and tracheal intubation time as MAC in the tongue edema scenario, but this was not statistically significant. The difficulty of using AVL was significantly lower than that of CML (8.8 vs. 3.3; *p* < 0.001). AVL may be an alternative for VL.

## 1. Introduction

Endotracheal intubation (ETI) is the insertion of a flexible plastic tube into the trachea. ETI is an essential emergency treatment to secure an airway in critically severe patients with loss of consciousness and respiratory failure, but it has been reported that it can also cause severe hypoxia (26%), hemodynamic collapse (25%), and cardiac arrest (2%) [1,2]. In emergency situations where time is insufficient for patient pretreatment and for gathering information about the patient, complications occurring after emergency ETI are bound to increase [3]. An investigation of emergency ETI involving 3423 cases over 7 years treated outside the operating room included difficult airways in 10.3% of cases, and composite complications in 4.2% of cases. Complications were observed and were related to aspiration (2.8%), esophageal intubation (1.3%), dental injury (0.2%), and pneumothorax (0.1%) [4].

Medical staff performed ETI directly using a Macintosh laryngoscope. However, to achieve a success rate of over 90%, approximately 47 to 56 times more experience is required, and even if you have a lot of experience, it cannot be easy to succeed in ETI in difficult airway situations [5]. Among all patients requiring ETI, the proportion of difficult intubations is said to range from approximately 6% to 30%, although there are wide range of variations depending on the literature [6,7].

The video laryngoscope (VL) was first introduced in 2001, representing a significant advance in tracheal intubation that was comparable to the invention of the Macintosh laryngoscope nearly 80 years ago [8]. VL has recently developed airway management by helping to overcome the difficulties in achieving adequate glottic visualization through direct laryngoscopy. The VL includes a laryngoscope with a high-quality digital camera mounted a few centimeters from the tip of the blade. This involves transferring images from this camera to a screen, providing an anterior view of the glottis and a wider viewing angle, thereby improving laryngeal visibility [9,10]. The use of this method is associated with improved glottic visibility, reduced airway trauma, and faster first intubation, especially for novice laryngoscopists, offering a safer risk profile compared to direct laryngoscopy [11]. Therefore, video laryngoscopy is increasingly recognized as an essential tool during intubation [12]. However, the learning curve involved in using a video laryngoscope correctly and developing the skills to maximize the likelihood of successful primary intubation still requires an adjustment during training. Doglioni et al. reported that the learning curve of clinicians performing neonatal intubation with direct laryngoscopy has not been fully described but may require more than 100 attempts to become proficient for primary intubation [13], and Uchinami et al. reported that the learning curve for VL compared to direct laryngoscopy was potentially shorter [14].

To minimize these drawbacks, we devised an attachable video laryngoscope (AVL) by the simple attachment of a camera and a monitor to a conventional Macintosh laryngoscope (CML). This device was developed through the technical guidance from medical doctors in the emergency medicine department, and most medical staff have a similar experience when using the CML. This point could emphasize the ease of access to the product, thus ensuring patient safety and staff satisfaction (Figure 1).

The purpose of this study was to assess the usefulness of the AVL in difficult airways. We hypothesized that this novel device would be more successful and easier to use than other VLs for physicians without VL experience but with CML experience.

## 2. Materials and Methods

### 2.1. Scenario Simulation

This study was designed as a comparative case–control study to evaluate tracheal intubation in normal airway and tongue edema scenarios. Also, the simulation was performed as a comparative evaluation study of various VLs using a manikin model for ETI (Deluxe Difficult Airway Trainer^®^; Laerdal, Stavanger, Norway). The manikin has a manually inflatable tongue to simulate obstructed airways or tongue edema. The manikin was placed at Wonju Severance Christian Hospital as a general specification. Therefore, the simulation scenarios could be adopted as normal airway and tongue edema.

### 2.2. Recruitment of Participants

The study participants were recruited from Wonju Severance Christian Hospital. Among the physicians who worked in the university hospital, 20 with no VL experience were recruited. The sample size was calculated using the G*Power 3.1.9.7 software [15,16]. An effect size of 1.33 was assumed, with an alpha level (α) of 0.05 and a power (1-β) of 0.8. Based on these parameters, the minimum required sample size for each group was calculated to be 10 participants. Of these, 10 physicians who had clinical experience in using tracheal intubation (skilled group), and another 10 physicians who were affiliated with other departments had little clinical experience using tracheal intubation (unskilled group). This study was also carried out with approval from the Research Ethics Committee of Yonsei University Wonju Severance Christian Hospital (IRB approval number: YWMR-13-02-024; approval date: 16 August 2013).

### 2.3. Equipment

All participants were briefed on the use of each device for 3 min prior to use. There was no rehearsal. The sequence of use of the devices was decided randomly by picking labeled pieces of paper whenever the scenario changed. The intent was to prevent learning effects, regency effects, and primacy effects. For tracheal intubation, a 7.5 mm diameter tube was used in all circumstances. The tube cuff was blocked using a 10 mL syringe. A resuscitator bag (Ambu, Copenhagen, Denmark) was used for ventilation. The laryngoscopes that were used are shown in Figure 2: ① McGrath MAC^®^ (Aircraft Medical Limited, Edinburgh, Scotland), ② AV-scope^®^ AVL (Caretek Inc., Wonju, Korea), ③ Blades 3 CML (Welch Allyn Inc., Skaneateles, NY, USA), and ④ Pentax Airwayscope^®^ (Pentax Corporation, Tokyo, Japan).

### 2.4. Data Collection

Tracheal intubation time was measured in seconds to the first decimal place using the smartphone stopwatch function until the tube was intubated. After ETI was performed, it was considered successful if breathing was confirmed using a positive-pressure ventilation indicator attached to the manikin. Also, tracheal intubation was considered to have failed if the tube was inserted into the esophagus resulting in gastric distension, or if the intubation time exceeded 30 s. The criterion of 30 s for tracheal intubation was set by the standards defined in the Advanced Cardiac Life Support [17]. Glottic view assessment was determined by a physician intubating the trachea with a laryngoscope, grading it using the Cormack and Lehane classification (Figure 3), and then inserting the tube [18]. After the experiment was completed, participants answered a questionnaire, using the Visual Analog Scale (VAS) to measure the degree of pain, on the ease of use by selecting the degree of difficulty of intubation from 0 (very easy) to 10 (very difficult).

### 2.5. Outcomes of Treatment

The primary outcomes of the study focused on the success rate and time required for tracheal intubation using various devices in normal airway and tongue edema scenarios. These results provided important insights into the efficacy and efficiency of AVL compared to CML and other video laryngoscopes (AWS and MAC). Secondary outcomes included an assessment of the degree of difficulty in using the device and the quality of glottic view as assessed by participating physicians. These comprehensive outcome measures highlight the potential benefits and utility of AVL as an alternative for airway management.

### 2.6. Statistical Analyses

Statistical analysis was performed using IBM SPSS Statistics Version 27.0 (IBM Corp., Armonk, NY, USA). The measured data for each group were expressed as frequency and percentage (%) for nominal and ordinal variables, and for quantitative variables, mean and standard deviation (SD) for normal distribution and median and interquartile range (IQR) for non-normal distribution. Since the values of success rate and glottic view assessment were nominal variables and a comparison was performed between the four types of laryngoscopes, the data for the success of the tracheal intubation attempts were analyzed using the chi-square test when they were nominal variables, and the data for the quality of the glottic view were analyzed using the Kruskal–Wallis test when they were ordinal variables. Data for intubation time and the degree of difficulty of intubation were tested for normal distribution using the Kolmogorov–Smirnov test. If data had a normal distribution, analysis of variance (ANOVA) was conducted. It was said to be significant if the *p*-value was less than 0.05.

## 3. Results

With two scenarios and four devices, 160 study cases were collected from the 20 participants.

### 3.1. Comparative Assessment of Device in the Skilled Group

Table 1 presents the comparative assessment of success rate, intubation time, visual field, and degree of difficulty of use of the device in the skilled group.

In the normal airway scenario, there were no significant differences in the success rates of tracheal intubation and intubation times with any of the devices (*p* = 0.38 and *p* = 0.68, respectively). The glottic view was classified as grade I-II, which indicated that it was possible to perform tracheal intubation with all devices.

In the tongue edema scenario, the Pentax AWS^®^ (AWS) had the highest success rate (100%) followed by the AVL (60%), McGrath MAC^®^ (MAC) (60%), and CML (20%) (*p* = 0.001). The intubation time in CML was the shortest (9.3 s). Also, the intubation time in AVL (19.2 [14.4–23.4] seconds) was longer than that of AWS (10.2 [9.8–11.7] seconds) but was similar to that of MAC (20.4 [16.6–23.9]) (*p* = 0.007).

Considering the degree of difficulty of use, AWS was the easiest to use by a significant degree (1.5 [0.4–2.6]). Although the difficulty of use of AVL (2.8 [2.1–4.3]) was more difficult than that of AWS, it was significantly easier to use compared to CML (vs. 7.8 [5.5–8.1]) and MAC (vs. 5.3 [3.9–5.8]) (*p* < 0.001).

### 3.2. Comparative Assessment of Device in the Unskilled Group

Table 2 presents the comparative assessment of success rate, intubation time, visual field, and degree of difficulty of use of the device in the unskilled group.

In the normal airway scenario, there were no significant differences in the success rates and times of tracheal intubation with any of the devices (*p* = 0.19 and *p* = 0.90, respectively). The glottic view was classified as grade I-II, which indicated that it was possible to perform tracheal intubation with all devices.

In the tongue edema scenario, the AWS had the highest success rate (80%) followed by the MAC (60%), AVL (50%), and CML (10%), but there were no significant differences (*p* = 0.12). The intubation time in AWS was the shortest (17.2 ± 11.9 s). Also, the intubation time in AVL (23.3 [18.8–27.3] seconds) was similar to that of MAC (27.2 ± 22.7 s) and CML (26.7 s) (*p* = 0.13).

Considering the degree of difficulty of use, the AWS was the easiest to use by a significant degree (0.7 [0.1–1.8]). Although the difficulty of use of AVL (3.3 [3.1–4.9]) was more difficult than that of AWS, it was significantly easier to use compared to CML (vs. 8.8 [6.9–9.1]) (*p* < 0.001).

## 4. Discussion

This study aimed to evaluate the usefulness of AVL in the simulated tracheal intubation scenario using a manikin. There was no difference in the tracheal intubation success rate and the intubation time in the normal airway scenario in both the skilled and unskilled groups. In the experienced group, AWS had the highest success rate in the tongue edema airway scenario, followed by AVL, MAC, and CML. The time required to intubate using AWS was significantly shorter than AVL or MAC. The difficulty of using AVL was lower than that of CML. For the experienced group, AVL was interpreted as being inferior to AWS but better than MAC. In the unskilled group, AVL had a similar success rate and tracheal intubation time as MAC in the tongue edema scenario, but this was not statistically significant. Also, the difficulty of using AVL was lower than that of CML. To manage patient safety and avoid ethical friction, it was thought that it would be most appropriate to use a manikin in this study. Additionally, the use of a manikin allowed the more precise control of the study conditions [19,20]. Comparisons of VL have used parameters including success rate, time of tracheal intubation, scale of the secured visual field (based on the Cormack and Lehane classification), and VAS-assessed degree of difficulty of use [19,21,22,23].

Direct laryngoscope has traditionally been the go to method for airway management in emergency departments, but VL has seen a consistent rise in use over the past decade. As recent as 2012, physicians were using direct laryngoscopes rather than VL in emergency rooms [24]. However, VL has emerged as a valuable tool in airway management, offering several advantages over traditional direct laryngoscopy. A study by Prekker et al. highlights its effectiveness in improving glottic visualization, particularly in patients with difficult airways or anatomical challenges [25]. The enhanced visualization provided by video laryngoscopes can lead to higher first-pass success rates and reduced intubation-related complications [26]. Additionally, video laryngoscopy allows for real-time feedback and teaching opportunities, aiding in the training of medical professionals [27]. Especially, the attachable video laryngoscope (AVL) developed in this study has a similar feeling of use because it attaches a camera and a monitor to the conventional Macintosh laryngoscope (CML) used by most medical physicians. Because this device is inexpensive and requires no special training or skills to use, it could be introduced into primary healthcare facilities in developing countries [28]. However, despite these benefits, video laryngoscopy also presents some drawbacks. Furthermore, the reliance on electronic equipment introduces the risk of technical failure, potentially hindering rapid airway management in critical situations [29]. Overall, while video laryngoscopy offers significant advantages in airway management, clinicians must carefully weigh these benefits against the associated challenges to determine its optimal utilization in clinical practice.

In the normal airway scenario, there were both reports that the VL was not useful for ETI in medical students who were not proficient in it and that it was useful [21,23]. Similar to the results of this study, there was no statistically significant difference for ETI with VL in the normal airway. On the other hand, in the tongue edema scenario, it was confirmed that the ETI success rate was higher than that of CML not only in AVL but also in other VLs such as AWS and MAC. Interestingly, in the unskilled group who were not familiar with the CML, the degree of difficulty in using the AVL was not appreciably different compared with the MAC, whereas the skilled group who were familiar with the CML did display an appreciable difference in comparison with the MAC. These outcomes were expected because the AVL has the same shape and is used in the same manner as the CML. Additionally, Yong et al. reported that as intubators gained experience, they showed less improvement when using a VL than when using a CML compared to actual novices [30].

Results have been reported from comparative studies on various video laryngoscopes. In the present study, the Pentax AWS^®^ was more functional than the other two VLs concerning outcomes and the degree of difficulty of use. The Pentax AWS^®^ was typically the easiest method concerning outcomes and the degree of difficulty of use. Similarly, it was reported to emphasize the superior glottic visualization provided by the Pentax AWS compared to direct laryngoscopy, particularly in patients with limited neck mobility or difficult airway anatomy [31]. These benefits should be considered in future AVL refinements. Additionally, its ergonomic handle design and intuitive user interface contribute to user satisfaction and the ease of use in various clinical settings [32]. The Pentax AWS^®^ differs from the conventional video laryngoscopes in that it assists with inserting the tracheal tube using a tube guide and a monitor to indicate the direction of tube progression [33]. Overall, the Pentax AWS stands out as a valuable tool in airway management, offering improved visualization, efficiency, and user-friendly features that benefit both patients and healthcare providers.

The VL represents the latest innovation in the area of airway management, enabling video-guided indirect laryngoscopy, and it is particularly useful for difficult airways. However, experience using the equipment and frequent use are required to ensure successful tracheal intubation [34,35]. Thus, the ideal device would be a tool that is designed to be used as a standard laryngoscope but that provides improved views of the larynx in an unanticipated difficult airway [36]. In this study, it was confirmed that AVL was competitive among VLs in terms of the degree of difficulty of use. The skilled group responded that although AVL was more difficult to use than AWS, it was easier to use than MAC. This can be thought of as being reflected in the purpose of developing AVL in this study, which is to enable users to use AVL with a similar feeling as using CML.

This study used a manikin simulation, which has some limitations. In the actual setting, fogging caused by breathing, bleeding due to trauma, vomiting, and secretions can all adversely affect the VL camera lens, but these real-world factors may not be influential in comparisons of the different VLs because the conditions were common through all comparisons. Nevertheless, real-world conditions should be considered in future studies, particularly when comparing an AVL with a general laryngoscope that has no camera. A second limitation concerns comparing the four devices in only one difficult scenario. There are multiple types of difficult airways including tongue edema, facial trauma, cervical immobility, jaw trismus, and pharyngeal obstruction. Only tongue edema was studied here because it is considered the most difficult airway obstacle [37,38]. This minimized learning effect bias. Still, information on device performance in all airway scenarios would be ideal. As a third limitation, we did not assess injury complications. Oral and dental injuries are also important in terms of the clinical view, but these complications are not important compared with the intubation success rate or time from the perspective of risk versus benefit in emergency airway management. Thus, they were not considered in this study.

## 5. Conclusions

In the skilled group, the AVL had the shortest tracheal intubation time compared to other devices in normal airway scenarios. Additionally, in the tongue edema scenario, the success rate of tracheal intubation was higher than that of CML, but the tracheal intubation time took longer. Similarly, in the unskilled group, the AVL showed a similar success rate and time of tracheal intubation to the MAC. The AVL may be an alternative for VL.

## Figures and Tables

**Figure 1 bioengineering-11-00570-f001:**
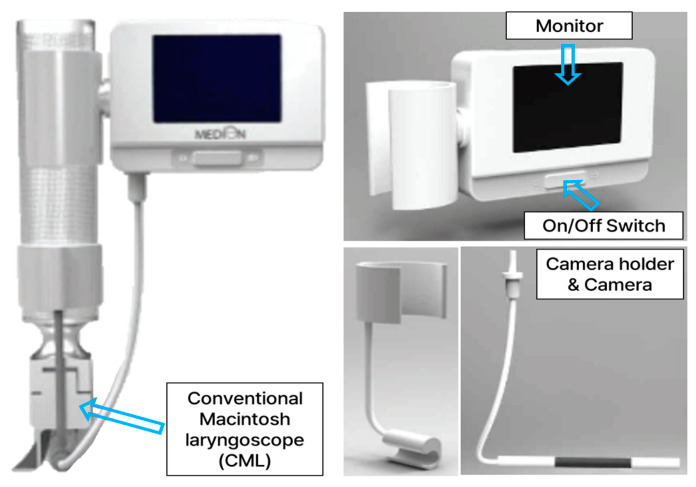
Photos of AVL including components.

**Figure 2 bioengineering-11-00570-f002:**
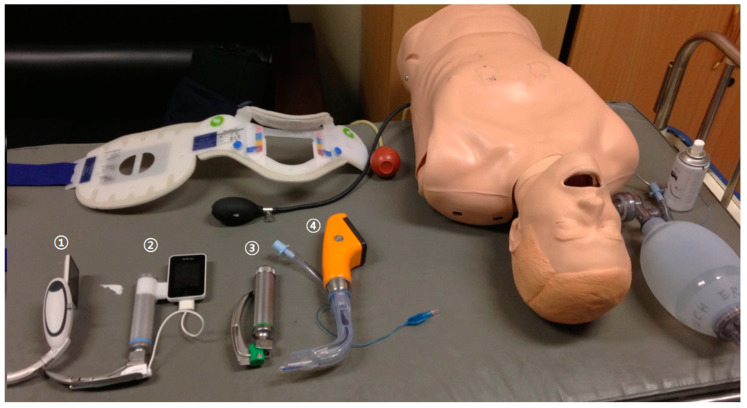
Equipment used in this study. The four laryngoscopes described in the text are numbered.

**Figure 3 bioengineering-11-00570-f003:**
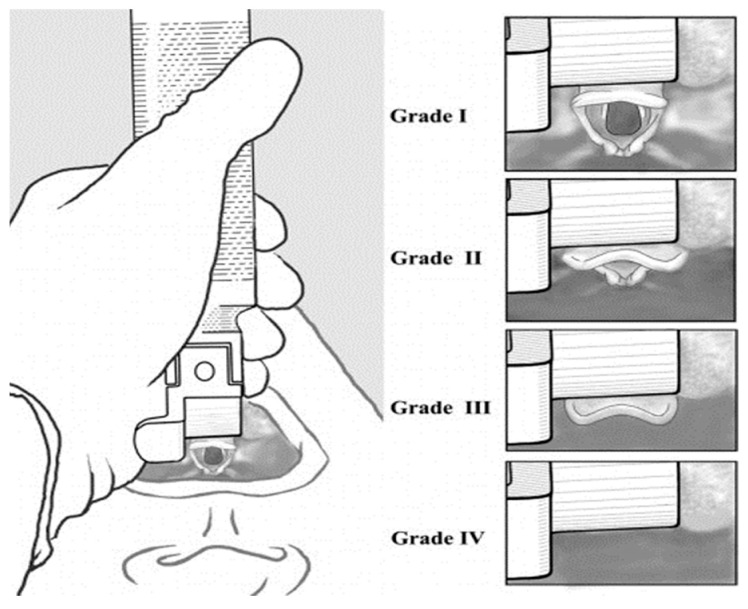
Cormack and Lehane Classification.

**Table 1 bioengineering-11-00570-t001:** Comparative assessment of success rate, intubation time, visual field, and degree of difficulty of use of device in the skilled group.

	AVL	CML	AWS	MAC	*p*-Value
Normal Airway					
Success Rate, n(%)	9(90)	10(100)	10(100)	10(100)	0.38
Tracheal Intubation Time, sec [M ± S.D *, IQR]	13.4 ± 5.5 *	15.0 [10.5–16.7]	11.6 [9.2–15.3]	16.1 ± 10.5 *	0.68
Grade of Glottic View, n(%)					<0.001
I	10(100)	3(30)	10(100)	8(80)	
II	0	7(70)	0	2(20)	
III	0	0	0	0	
IV	0	0	0	0	
Tongue Edema					
Success Rate, n(%)	6(60)	1(10)	10(100)	6(60)	0.001
Tracheal Intubation Time, sec [M ± S.D *, IQR]	19.2 [14.4–23.4]	9.3 [9.3–9.3]	10.2 [9.8–11.7]	20.4 [16.6–23.9]	0.007
Grade of Glottic View, n(%)					<0.001
I	8(80)	0	9(90)	7(70)	
II	2(20)	2(20)	0	1(10)	
III	0	3(30)	1(10)	1(10)	
IV	0	5(50)	0	1(10)	
Degree of Difficulty of Use [M ± S.D *, IQR]	2.8 [2.1–4.3]	7.8 [5.5–8.1]	1.5 [0.4–2.6]	5.3 [3.9–5.8]	<0.001

* Kolmogorov–Smirnov test (AVL—attachable video laryngoscope; CML—conventional Macintosh laryngoscope; AWS—Pentax Airwayscope; MAC—McGrath MAC; M—mean; S.D—standard deviation; and IQR—interquartile range).

**Table 2 bioengineering-11-00570-t002:** Comparative assessment of success rate, intubation time, visual field, and degree of difficulty of use of device in the unskilled group.

	AVL	CML	AWS	MAC	*p*-Value
Normal Airway					
Success Rate, n(%)	6(60)	5(50)	8(80)	9(90)	0.19
Tracheal Intubation Time, sec [M ± S.D *, IQR]	18.8 [11.4–21.8]	15.7 [12.5–24.5]	19.6 ± 7.7 *	13.4 [10.9–17.9]	0.90
Grade of Glottic View, n(%)					<0.001
I	10(100)	3(30)	10(100)	9(90)	
II	0	7(70)	0	1(10)	
III	0	0	0	0	
IV	0	0	0	0	
Tongue Edema					
Success Rate, n(%)	4(40)	1(10)	8(80)	6(60)	0.12
Tracheal Intubation Time, sec [M ± S.D *, IQR]	23.3 [18.8–27.3]	26.7 [26.7–26.7]	17.2 ± 11.9 *	27.2 ± 22.7 *	0.13
Grade of Glottic View, n(%)					<0.001
I	10(100)	0	9(90)	8(80)	
II	0	2(20)	0	2(20)	
III	0	4(40)	1(10)	0	
IV	0	4(40)	0	0	
Degree of Difficulty of Use [M ± S.D *, IQR]	3.3 [3.1–4.9]	8.8 [6.9–9.1]	0.7 [0.1–1.8]	2.4 [2.0–3.3]	<0.001

* Kolmogorov–Smirnov test (AVL—attachable video laryngoscope; CML—conventional Macintosh laryngoscope; AWS—Pentax Airwayscope; MAC—McGrath MAC; M—mean; S.D—standard deviation; and IQR—interquartile range).

## Data Availability

The data that support the findings of this study are available on request from the corresponding author. The data are not publicly available due to privacy or ethical restrictions.

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
