# Peer review of "The Clinical Usability Evaluation of an Attachable Video Laryngoscope in the Simulated Tracheal Intubation Scenario: A Manikin Study"

_bioengineering, 2024, doi:10.3390/bioengineering11060570_

Round 1

Reviewer 1 Report

Comments and Suggestions for Authors

The authors have presented a study evaluating 4 different tools in two different clinical situations in two groups of physicians (skilled and unskilled) to evaluate whether AVL is equivalent to Pentax AWS.

It is a simple descriptive study 

Abstract: Please add the full form of AWS  - airway scope

Methodology: It is useful to stratify physicians as senior, or junior based on their experience and see if there are differences. More depth in the evaluation to simulate the different real-life clinical situations would have been useful

Discussion: It is useful to discuss the cost of the equipment

Comments on the Quality of English Language

Grammar and language could be improved

Line 42: grammar (and observed)

Author Response

※ Comments and Suggestions for Authors

The authors have presented a study evaluating 4 different tools in two different clinical situations in two groups of physicians (skilled and unskilled) to evaluate whether AVL is equivalent to Pentax AWS.

It is a simple descriptive study 

Abstract: Please add the full form of AWS  - airway scope

 â˜ž Thank you for your comment. We wrote the full name of Pentax AWS according to your advice.

☞ [Before] (Line 13) ~ CML, AVL, Pentax AWS® (AWS), and McGrath MAC® (MAC) ~

   [Revised] (Line 13) ~ CML, AVL, Pentax Airwayscope® (AWS), and McGrath MAC® (MAC) ~

Methodology: It is useful to stratify physicians as senior, or junior based on their experience and see if there are differences. More depth in the evaluation to simulate the different real-life clinical situations would have been useful

☞ Thank you for your sincere opinion. As you mentioned, we fully understood that performing the clinical trial would have been more useful by making a group of senior and junior physicians based on experience. In this study, we focused on whether or not there was video laryngoscopy experience, so we decided to make two groups by department, so we thought about dividing it into emergency medicine and other departments. In further research, we will try to design a more detailed study by referring to your advice.

Discussion: It is useful to discuss the cost of the equipment

☞ Thank you for your advice. According to your mention, we added the sentences related to the cost of the equipment and revised the paragraph.

☞ [Before] (Line 178-184) Additionally, video laryngoscopy allows for real-time feedback and teaching opportunities, aiding in the training of medical professionals [23]. However, despite these benefits, video laryngoscopy also presents some drawbacks. Cost implications, device availability, and the learning curve associated with its use are notable concerns [22]. Furthermore, the reliance on electronic equipment introduces the risk of technical failure, potentially hindering rapid airway management in critical situations [24].

   [Revised] (Line 178-187) Additionally, video laryngoscopy allows for real-time feedback and teaching opportunities, aiding in the training of medical professionals [23]. Especially, the attachable video laryngoscope (AVL) developed in this study has a similar feeling of use because it attaches a camera and monitor to the conventional Macintosh laryngoscope (CML) used by most medical physicians. Because this device is inexpensive and requires no special training or skills to use, it could be introduced into primary healthcare facilities in developing countries [24]. However, despite these benefits, video laryngoscopy also presents some drawbacks. Cost implications, device availability, and the learning curve associated with its use are notable concerns [22]. Furthermore, the reliance on electronic equipment introduces the risk of technical failure, potentially hindering rapid airway management in critical situations [25].

※ Comments on the Quality of English Language

Grammar and language could be improved

Line 42: grammar (and observed)

☞ Thank you for pointing out the lack of grammar. We tried to improve the grammar and rewrote the paragraph correctly.

☞ [Before] (Line 39-42) An investigation of emergency ETI involving 3,423 cases over 7 years treated outside the operating room revealed difficult airways in 10.3% of cases, with 4.2% composite complications among 2,294 senior residents involving aspiration (3.0%), esophageal intubation (1.1%), dental injury (0.2%), and pneumothorax (0.04%) and observed [4].

   [Revised] (Line 39-42) An investigation of emergency ETI involving 3,423 cases over 7 years treated outside the operating room included difficult airways in 10.3% of cases, and with 4.2% composite complications. Complications were observed in aspiration (2.8%), esophageal intubation (1.3%), dental injury (0.2%), and pneumothorax (0.1%) [4].

Reviewer 2 Report

Comments and Suggestions for Authors

The author evaluated the clinical usability of an add-on video laryngoscope in a simulated tracheal intubation scenario. The success rate, time and ease of intubation of four kinds of laryngoscope were compared by simulating normal and tongue edema airway scenarios. Based on the consideration of the whole paper, it is recommended to publish it in “Bioengineering” after revising the following questions. The comments are as follows:

1. It is recommended that the authors more explicitly describe the details of the study design in the article, including how the normal and tongue edema airway scenarios were simulated, how the physicians participating in the study were selected, and how they were assigned to the skilled and unskilled groups.

2. Please provide more information on data collection and analysis, including how intubation time was recorded, success rates, use of difficulty scores, and specific methods for laryngeal view evaluation.

3. While AWS, AVL, MAC, and CML were all used for comparison, it is recommended that the authors further explain why these devices were selected as the control group and discuss the impact of these choices on the study results.

4. In the discussion section, it is suggested that the author explain the study results in more depth, especially the reasons for the high success rate of intubation in the tongue edema scenario of AWS, and the advantages of AVL compared with CML in the difficulty of use. In addition, the possible implications of these results for clinical practice can also be explored.

5. A variety of statistical tests are mentioned in the paper, but the data preprocessing and the selection of statistical models are not explained in detail. It is recommended that the authors provide more details on data normality testing, outlier handling, and model selection.

Author Response

Reviewer: 2

※ Comments and Suggestions for Authors

The author evaluated the clinical usability of an add-on video laryngoscope in a simulated tracheal intubation scenario. The success rate, time and ease of intubation of four kinds of laryngoscope were compared by simulating normal and tongue edema airway scenarios. Based on the consideration of the whole paper, it is recommended to publish it in “Bioengineering” after revising the following questions. The comments are as follows:

  1. It is recommended that the authors more explicitly describe the details of the study design in the article, including how the normal and tongue edema airway scenarios were simulated, how the physicians participating in the study were selected, and how they were assigned to the skilled and unskilled groups.

☞ Thank you for your opinion. We added sentences about the process and reasons for adopting the scenario.

☞ [Before] (Line 75-78) The simulation was performed as a comparative evaluation study of various VLs using a manikin model for ETI (Deluxe Difficult Airway Trainer®; Laerdal, Stavanger, Norway). The manikin was placed on a hospital bed with a general specification. The simulation scenarios included a normal airway and tongue edema.

[Revised] (Line 75-80) The simulation was performed as a comparative evaluation study of various VLs using a manikin model for ETI (Deluxe Difficult Airway Trainer®; Laerdal, Stavanger, Norway). The manikin has a manually inflatable tongue to simulate obstructed airway or tongue edema. The manikin used to be placed at Wonju Severance Christian Hospital as a general specification. Therefore, the simulation scenarios could be adopted as normal airway and tongue edema.

  1. Please provide more information on data collection and analysis, including how intubation time was recorded, success rates, use of difficulty scores, and specific methods for laryngeal view evaluation.

☞ Thank you for your comments. We rewrote the paragraph to provide more information on data collection and analysis.

☞ [Before] (Line 102-111) When the respiration was confirmed by a positive pressure ventilation indicator attached to the manikin, tracheal intubation was considered successful. When the tube was inserted into the esophagus and caused stomach expansion, or when the intubation time exceeded 30 seconds, tracheal intubation was considered to have failed. The criterion of 30 seconds for tracheal intubation was set by the standards defined in the Advanced Cardiac Life Support [13]. The glottic view assessment was decided by the physician intubating the trachea with a laryngoscope, determining the grade using the Cormack and Lehane classification (Figure 3), and then inserting the tube [14]. The degree of difficulty of intubation according to the participants was ranked using a Visual Analog Scale (VAS) that ranged from 0 (very easy) to 10 (very difficult) after study completion.

   [Revised] (Line 102-114) Tracheal intubation time was measured in seconds to the first decimal place using the smartphone stopwatch function until the tube was intubated. After ETI was performed, it was considered successful if breathing was confirmed using a positive pressure ventilation indicator attached to the manikin. Also, tracheal intubation was considered failed if the tube was inserted into the esophagus resulting in gastric distension, or if the intubation time exceeded 30 seconds. The criterion of 30 seconds for tracheal intubation was set by the standards defined in the Advanced Cardiac Life Support [13]. Glottic view assessment was determined by a physician intubating the trachea with a laryngoscope, grading it using the Cormack and Lehane classification (Figure 3), and then inserting the tube [14]. After the experiment was completed, participants conducted a questionnaire, using the Visual Analog Scale (VAS) used to usually measure the degree of pain, on ease of use by selecting the degree of difficulty of intubation from 0 (very easy) to 10 (very difficult).

  1. While AWS, AVL, MAC, and CML were all used for comparison, it is recommended that the authors further explain why these devices were selected as the control group and discuss the impact of these choices on the study results.

☞ Thank you for your curiosity and meaningful comments. The attachable video laryngoscope (AVL), which we developed in this study, is an improved laryngoscope equipped with a monitor and camera on the conventional Macintosh laryngoscope (CML) to maintain the same sensation as using the existing CML <line 63-66>. Basically, we designed a study to compare AVL and CML, and chose AWS and MAC as controls to make meaningful comparisons between VLs as well. Through the results of this study, we tried to present the sentences in DISCUSSION section related to MAC and AWS as follows. AWS is known to have the best usability and convenience in VL, so it was described as a standard for comparison (lines 197-208), and MAC has a similar appearance to CML, but is integrated, so it was expected that there would be differences when compared to AVL. (lines 187-194).

☞ (Line 190-194) ~~~. Interestingly, in the unskilled group who were not familiar with the CML, the de-gree of difficulty in using the AVL was not appreciably different compared with the McGrath MAC®, whereas the skilled group who were familiar with the CML did display an appreciable difference in comparison with the McGrath MAC®. These outcomes were expected because the AVL has the same shape and is used in the same manner as the CML.

(Line 203-208) The Pentax AWS® differs from the conventional video laryngoscopes in that it assists with inserting the tracheal tube using a tube guide and a monitor to indicate the direction of tube progression [27]. Overall, the Pentax AWS stands out as a valuable tool in airway management, offering improved visualization, efficiency, and user-friendly features that benefit both patients and healthcare providers.

  1. In the discussion section, it is suggested that the author explain the study results in more depth, especially the reasons for the high success rate of intubation in the tongue edema scenario of AWS, and the advantages of AVL compared with CML in the difficulty of use. In addition, the possible implications of these results for clinical practice can also be explored.

☞ Thank you for your comments. We tried to revise the sentence to describe the advantages of AVL in more detail.

☞ [Before]

(Line 187-194) In the normal airway scenario, the choice of the device was not influential to the outcome. In the normal airway setting, prior findings have been equivocal, with VL being found use and not useful in college students who were unskilled at tracheal intubation [17,18]. Interestingly, in the unskilled group who were not familiar with the CML, the degree of difficulty in using the AVL was not appreciably different compared with the McGrath MAC®, whereas the skilled group who were familiar with the CML did display an appreciable difference in comparison with the McGrath MAC®. These outcomes were expected because the AVL has the same shape and is used in the same manner as the CML.

(Line 209-215) The VL represents the latest innovation in the area of airway management, enabling video-guided indirect laryngoscopy, and it is particularly useful for difficult airways. However, experience using the equipment and frequent use is required to ensure successful tracheal intubation [28,29]. Thus, the ideal device would be a tool that is designed to be used as a standard laryngoscope but that provides improved views of the larynx in an unanticipated difficult airway [30]. However, in the results obtained from this study, the AVL did not exhibit usefulness compared to the other two VLs.

   [Revised]

(Line 203-213) In the normal airway scenario, there were both reports that the VL was not useful for ETI in medical students who were not proficient in it, and that it was useful [17, 19]. Similar to the results of this study, there was no statistically significant difference for ETI with VL in the normal airway. On the other hand, in the tongue edema scenario, it was confirmed that the ETI success rate was higher than that of CML not only in AVL but also in other VLs such as AWS and MAC. Interestingly, in the unskilled group who were not familiar with the CML, the degree of difficulty in using the AVL was not appreciably different compared with the MAC, whereas the skilled group who were familiar with the CML did display an appreciable difference in comparison with the MAC. These outcomes were expected because the AVL has the same shape and is used in the same manner as the CML.

(Line 225-234) The VL represents the latest innovation in the area of airway management, enabling video-guided indirect laryngoscopy, and it is particularly useful for difficult airways. However, experience using the equipment and frequent use is required to ensure successful tracheal intubation [29,30]. Thus, the ideal device would be a tool that is designed to be used as a standard laryngoscope but that provides improved views of the larynx in an unanticipated difficult airway [31]. In this study, it was confirmed that AVL was competitive among VLs in terms of the degree of difficulty of use. The skilled group responded that although AVL was more difficult to use than AWS, it was easier to use than MAC. This can be thought of as being reflected in the purpose of developing AVL in this study, which is to enable users to use AVL with a similar feeling to using CML.

  1. A variety of statistical tests are mentioned in the paper, but the data preprocessing and the selection of statistical models are not explained in detail. It is recommended that the authors provide more details on data normality testing, outlier handling, and model selection.

☞ Thank you for pointing out the lack of description of our statistical analysis methods. Following your comments, we tried to describe in more detail about handling data. We rewrote the paragraph on statistical analysis methods.

☞ [Before] (Line 114-121) Statistical analysis was performed using IBM SPSS Statistics Version 27.0 (IBM Corp., Armonk, NY, USA). The data for the success of the tracheal intubation attempts were analyzed using the χ2 test. Data for the quality of the glottic view were analyzed using the Kruskal-Wallis test. Data for intubation time and the degree of difficulty of intubation were tested for normal distribution using Kolmogorov-Smirnov test. If data had a normal distribution, analysis of variance (ANOVA) was conducted. If data were not normally distributed, the Kruskal-Wallis test with Bonferroni correction was used. The α-error level for all analyses was set at P < 0.05.

   [Revised] (Line 119-131) Statistical analysis was performed using IBM SPSS Statistics Version 27.0 (IBM Corp., Armonk, NY, USA). The measured data for each group was expressed as frequency and percentage (%) for nominal and ordinal variables, and for quantitative variables, mean and standard deviation (SD) for normal distribution and median and interquartile range (IQR) for non-normal distribution. Since the values of success rate and glottic view assessment were nominal variables and a comparison between four types of laryngoscopes, the data for the success of the tracheal intubation attempts were analyzed using the chi-square test when they were nominal variables, and the data for the quality of the glottic view were analyzed using the Kruskal-Wallis test when they were ordinal variables. Data for intubation time and the degree of difficulty of intubation were tested for normal distribution using the Kolmogorov-Smirnov test. If data had a normal distribution, analysis of variance (ANOVA) was conducted. It was said to be significant if the p-value was less than 0.05.

Reviewer 3 Report

Comments and Suggestions for Authors

The authors investigated the usefulness of an attachable video laryngoscope (AVL), in which a camera and monitor are attached to a conventional Macintosh laryngoscope (CML).

The manuscript is well organized and interesting, but several significant methodological flaws in the study design were noted that should be clarified or revised. My comments and suggestions for improvement are as follows:

1. The abstract should be divided into chapters or the authors should reword the first sentence. They started the abstract with: "To evaluate the benefits of...". The authors should add: "The aim of this study was ....". Please revise

2. Introduction – The authors state that the learning curve for correct use of a video laryngoscope and developing the skills to maximize the likelihood of successful primary intubation still requires a learning curve that needs to be adapted during training. I fully agree with this, but the authors should add some data on the number of intubations required to reach a plateau on the learning curve and also provide a reference to support this statement.

3. In addition, the authors should add a few lines in the introduction about the available laryngoscopes and describe the differences between them in more detail.

4. The authors state that the study protocol was approved by the Research Ethics Committee of Yonsei University Wonju Severance Christian Hospital (IRB approval number: 86YWMR-13-02-024). Please add the date of approval next to the approval number.

5. The main methodological objection to this study is the lack of sample size calculation. As this is a prospective study, sample size calculation is mandatory. Please comment and clarify this point! If this was not done, the authors should clearly state why and add this to the study limitations. If a sample size calculation was performed, please include this as a separate paragraph in the methodology

6. Please add a new paragraph in methodology (Outcomes of treatment) and describe the primary and secondary outcomes of the study.

7. The authors state that they tested their data for normal distribution using the Kolmogorov-Smirnov test. In the results section and in the tables, it is obvious that the author reported all continuous variables with median and interquartile range (IQR). Normally distributed data should be reported as mean and standard deviation (SD), while non-normally distributed data should be reported as median (IQR). Please revise!

8. Any abbreviation given in a table should be explained in a table legend. Please revise all tables.

9. Tables – Next to each p-value, the authors should indicate which statistical test was used. In addition, the term [quartile] should be replaced by [IQR].

10. Although the discussion is well organized, the authors should state the main findings of their study in a few sentences in the first paragraph of the discussion (without results/ numbers).

11. The majority of references are outdated, there are only 2-3 references from the last 5 years. Please provide better search of literature search and the inclusion of more recent references.

Comments on the Quality of English Language

Minor editing of English language required

Author Response

Reviewer: 3

※ Comments and Suggestions for Authors

The authors investigated the usefulness of an attachable video laryngoscope (AVL), in which a camera and monitor are attached to a conventional Macintosh laryngoscope (CML).

The manuscript is well organized and interesting, but several significant methodological flaws in the study design were noted that should be clarified or revised. My comments and suggestions for improvement are as follows:

  1. The abstract should be divided into chapters or the authors should reword the first sentence. They started the abstract with: "To evaluate the benefits of...". The authors should add: "The aim of this study was ....". Please revise

☞ Thank you for your kind advice. We revised the sentence according to your comment.

☞ [Before] (Line 10-11) To assess the usefulness of an attachable video laryngoscope (AVL) by attaching a camera and monitor to a conventional Macintosh laryngoscope (CML).

   [Revised] (Line 10-11) The aim of this study was to assess the usefulness of an attachable video laryngoscope (AVL) by attaching a camera and monitor to a conventional Macintosh laryngoscope (CML).

  1. Introduction – The authors state that the learning curve for correct use of a video laryngoscope and developing the skills to maximize the likelihood of successful primary intubation still requires a learning curve that needs to be adapted during training. I fully agree with this, but the authors should add some data on the number of intubations required to reach a plateau on the learning curve and also provide a reference to support this statement.

☞ Thank you for your good opinion. We added the sentences related to the learning curve for the use of a video laryngoscope to perform primary intubation.

☞ [Before] (Line 60-62) ~~ However, the learning curve involved in using a video laryngoscope correctly and developing the skills to maximize the likelihood of successful primary intubation still requires a learning curve that must be adjusted during training.

   [Revised] (Line 59-65) ~~ However, the learning curve involved in using a video laryngoscope correctly and developing the skills to maximize the likelihood of successful primary intubation still requires a learning curve that must be adjusted during training. Doglioni et al. reported that the learning curve of clinicians performing neonatal intubation with direct laryngoscopy has not been fully described but may require more than 100 attempts to become proficient for primary intubation [13], and Uchinami et al. reported that the learning curve for VL compared to direct laryngoscopy was potentially shorter [14].

  1. In addition, the authors should add a few lines in the introduction about the available laryngoscopes and describe the differences between them in more detail.

☞ Thank you for your additional comments. We tried to explain the differences in more detail by highlighting the process of developing AVL and its ease of use.

☞ [Before] (Line 63-66) To minimize these drawbacks, we devised an attachable video laryngoscope (AVL) by the simple attachment of a camera and monitor to a conventional Macintosh laryngoscope (CML). Because most medical staff can adopt the feeling similarly of using CML, the product used in this study does not require specialized training or skills (Figure 1).

    [Revised] (Line 66-71) To minimize these drawbacks, we devised an attachable video laryngoscope (AVL) by the simple attachment of a camera and monitor to a conventional Macintosh laryngoscope (CML). This device was developed through technical guidance from medical doctors in the emergency medicine department, and most medical staff can feel similar to using the CML. This point could emphasize the ease of access to the product, thus ensuring patient safety and staff satisfaction (Figure 1).

  1. The authors state that the study protocol was approved by the Research Ethics Committee of Yonsei University Wonju Severance Christian Hospital (IRB approval number: 86YWMR-13-02-024). Please add the date of approval next to the approval number.

☞ Thank you for pointing out our mistake. We have listed the IRB approval date next to the IRB approval number.

☞ [Before] (Line 85-87) This study was also carried out with approval from the Research Ethics Committee of Yonsei University Wonju Severance Christian Hospital (IRB approval number: YWMR-13-02-024).

    [Revised] (Line 96-98) This study was also carried out with approval from the Research Ethics Committee of Yonsei University Wonju Severance Christian Hospital (IRB approval number: YWMR-13-02-024, Approval date: 16th August 2013).

  1. The main methodological objection to this study is the lack of sample size calculation. As this is a prospective study, sample size calculation is mandatory. Please comment and clarify this point! If this was not done, the authors should clearly state why and add this to the study limitations. If a sample size calculation was performed, please include this as a separate paragraph in the methodology

☞ Thank you for raising an important point. We completely agreed with you and we fully understood that this was a prospective study and sample size calculations were essential. However, unfortunately, at that time we had no comments regarding the sample size during the IRB review process at Wonju Severance Christian Hospital. When we announced that this was a mannequin study conducted on 20 doctors working at the hospital, dividing them into skilled and unskilled groups, we received IRB approval without a review opinion on the sample size. Nonetheless, following your comments, we tried to clarify the sample size calculation in the methodology.

☞ [Before] (Line 80-85) The study participants were recruited from Wonju Severance Christian Hospital. Among the physicians who worked in the university hospital, 20 with no VL experience were recruited. Of these, 10 physicians who had clinical experiences in emergent using tracheal intubation (skilled group), and another 10 physicians who were affiliated with other departments, had few clinical experiences using tracheal intubation (unskilled group).

   [Revised] (Line 88-96) The study participants were recruited from Wonju Severance Christian Hospital. Among the physicians who worked in the university hospital, 20 with no VL experience were recruited. The sample size was calculated using G*Power 3.1.9.7 software [15, 16]. An effect size of 1.33 was assumed, with an alpha level (α) of 0.05 and a power (1-β) of 0.8. Based on these parameters, the minimum required sample size for each group was calculated to be 10 participants. Of these, 10 physicians who had clinical experiences in emergent using tracheal intubation (skilled group), and another 10 physicians who were affiliated with other departments with few clinical experiences using tracheal intubation (unskilled group).

  1. Please add a new paragraph in methodology (Outcomes of treatment) and describe the primary and secondary outcomes of the study.

☞ Thank you for your comments for our better-quality manuscript. Following your advice, we added a new paragraph describing the primary and secondary outcomes of this study.

☞ [Revised] (Line 129-137)

2.5. Outcomes of treatment

The primary outcomes of the study focused on the success rate and time required for tracheal intubation using various devices in normal airway and tongue edema scenarios. These results provided important insights into the efficacy and efficiency of AVL compared to CML and other video laryngoscopes (AWS and MAC). Secondary outcomes included an assessment of the degree of difficulty in using the device and the quality of glottic view as assessed by participating physicians. These comprehensive outcome measures highlight the potential benefits and utility of AVL as an alternative option for airway management.

  1. The authors state that they tested their data for normal distribution using the Kolmogorov-Smirnov test. In the results section and in the tables, it is obvious that the author reported all continuous variables with median and interquartile range (IQR). Normally distributed data should be reported as mean and standard deviation (SD), while non-normally distributed data should be reported as median (IQR). Please revise!

☞ Thank you for pointing out the shortcomings in the statistics. We checked whether the distribution was normal or non-normal and revised the values in the table correctly. We also rewrote the results paragraph to reflect the revised values.

☞ [Before] (Table 1, Table 2)

[Revised] (Table 1, Table 2)

  1. Any abbreviation given in a table should be explained in a table legend. Please revise all tables.

☞ Thank you for pointing out our shortcomings. We wrote explanations of the abbreviations listed in the table below the table.

☞ [Revised] (Below Table 1, below Table 2) (AVL – attachable video laryngoscope, CML – conventional Macintosh laryngoscope, AWS – Pentax airwayscope, MAC – McGrath MAC, M – mean, S.D – standard deviation, IQR – interquartile range)

  1. Tables – Next to each p-value, the authors should indicate which statistical test was used. In addition, the term [quartile] should be replaced by [IQR].

☞ Thank you for pointing out our shortcomings. We changed the term [quartiles] to [IQR].

☞ [Before] (In Table 1, in Table 2) [quartiles]

     [Revised] (In Table 1, in Table 2) [IQR]

  1. Although the discussion is well organized, the authors should state the main findings of their study in a few sentences in the first paragraph of the discussion (without results/ numbers).

☞ Thank you for your opinion. We added a few sentences to describe the main findings of this study.

☞ [Before] (Line 163-169) This study aimed to evaluate the usefulness of AVL in the simulated tracheal intubation scenario using a manikin. To manage patient safety, and avoid ethical friction, it was thought that it would be most appropriate to use a manikin in this study. Additionally, the use of a manikin allowed more precise control of the study conditions [15,16]. Comparisons of VL have used parameters including success rate, time of tracheal intubation, scale of the secured visual field (based on the Cormack and Lehane classification), and VAS-assessed degree of difficulty of use [15,17,18,19]. 11. The majority of references are outdated, there are only 2-3 references from the last 5 years. Please provide better search of literature search and the inclusion of more recent references.

   [Revised] (Line 201-215) This study aimed to evaluate the usefulness of AVL in the simulated tracheal intubation scenario using a manikin. There was no difference in tracheal intubation success rate and intubation time in normal airway scenario in both skilled and unskilled groups. In the experienced group, AWS had the highest success rate in the tongue edema airway scenario, followed by AVL, MAC, and CML. The time required to intubate using AWS was significantly shorter than AVL or MAC. The difficulty of using AVL was lower than that of CML. For the experienced group, AVL was interpreted as being inferior to AWS but better than MAC. In the unskilled group, AVL had a similar success rate and tracheal intubation time as MAC in the tongue edema scenario, but this was not statistically significant. Also, the difficulty of using AVL was lower than that of CML. To manage patient safety, and avoid ethical friction, it was thought that it would be most appropriate to use a manikin in this study. Additionally, the use of a manikin allowed more precise control of the study conditions [17,18]. Comparisons of VL have used parameters including success rate, time of tracheal intubation, scale of the secured visual field (based on the Cormack and Lehane classification), and VAS-assessed degree of difficulty of use [19,21,22,23].

※ Comments on the Quality of English Language

Minor editing of English language required

Round 2

Reviewer 3 Report

Comments and Suggestions for Authors

The authors have adequately addressed my comments. In my opinion, the paper can be published in present form.

Comments on the Quality of English Language

Minor editing of English language required.